# Breaking the Dormancy of Snake’s Head Fritillary (*Fritillaria meleagris* L.) In Vitro Bulbs—Part 1: Effect of GA_3_, GA Inhibitors and Temperature on Fresh Weight, Sprouting and Sugar Content

**DOI:** 10.3390/plants9111449

**Published:** 2020-10-27

**Authors:** Marija Marković, Milana Trifunović Momčilov, Branka Uzelac, Aleksandar Cingel, Snežana Milošević, Slađana Jevremović, Angelina Subotić

**Affiliations:** Department of Plant Physiology, Institute for Biological Research “Siniša Stanković”-National Institute of Republic of Serbia, University of Belgrade, Bulevar Despota Stefana 142, 11060 Belgrade, Serbia; milanag@ibiss.bg.ac.rs (M.T.M.); branka@ibiss.bg.ac.rs (B.U.); cingel@ibiss.bg.ac.rs (A.C.); snezana@ibiss.bg.ac.rs (S.M.); sladja@ibiss.bg.ac.rs (S.J.); heroina@ibiss.bg.ac.rs (A.S.)

**Keywords:** *F. meleagris*, dormancy, sugars, chilling

## Abstract

Bulbs are the main vegetative reproductive organs of *Fritillaria meleagris* L. In nature, as well as in vitro, they become dormant and require low temperatures for further growth during the next vegetative period. In the present study, using 10 μM of gibberellic acid (GA_3_), or gibberellin biosynthesis (GA) inhibitors—ancymidol (A) and paclobutrazol (P)—the dynamic changes in soluble sugars, fructose and glucose content, fresh weight and sprouting capacity were investigated. *F. meleagris* bulbs were cultured on medium with GA_3_ and GA inhibitors for 1, 2 and 5 weeks at two different temperatures (24 and 7 °C). GA_3_ improved bulb fresh weight, as well as sprouting percentage at both tested temperatures, compared to the control. The highest fresh weight increase (57.7%) and sprouting rate (29.02%) were achieved when bulbs were grown at 24 °C for 5 weeks. In addition, soluble sugar content was the highest in bulbs grown for 5 weeks on medium supplemented with GA_3_. The main sugar in fritillary bulbs was glucose, while fructose content was lower. The sensitivity of bulbs to GA inhibitors differed and significantly affected sugar content in bulbs. To our knowledge, this is the first study of the sugar composition in *F. meleagris* bulbs during breaking of the bulb’s dormancy and its sprouting.

## 1. Introduction

Snake’s head fritillary *(Fritillaria meleagris* L.) belongs to the Liliaceae family, which are bulbous geophyte plants. *Fritillaria* species are mainly distributed throughout temperate climates of the Northern Hemisphere. These plants spend a particular period of the year in the form of dormant bulbs under the ground. The production of fritillaries with very attractive flowers, used for horticultural purposes, by conventional methods is very slow and it takes several years to produce a whole plant [1]. Beside horticultural production, these plants have potential for medical use, because of their phytochemical properties [2]. In vitro plant propagation can enhance the multiplication of this plant [2,3]. Normally, *F. meleagris* plants produce one single bulb in vitro, but bulb multiplication can be increased by various in vitro protocols and by the addition of different plant growth regulators in the culture medium. For successful in vitro bulb production, it was shown that many factors are important, but the most important are bulb size and its maturity [4,5]. Considering the abovementioned characteristics, researchers have focused on studying protocols for sprouting, storage temperature, plant growth, breaking dormancy and sprouting improvement.

Geophytes develop bulb dormancy to survive unfavorable environmental conditions [6]. For dormancy breaking, low temperature treatment (during a determined period of time) is very often required for normal plant development. If there is no low temperature period during the year, plant growth is very slow, and the flowers do not develop or the formed flowers are deformed [7]. Germination and breaking the dormancy and further plant growth depend on external factors and these physiological processes are stopped without proper external conditions in a manner that is not fully clarified [8].

Various physiological and morphological changes have been observed in the dormant bulbs and bulbs which begin to sprout after a period of low temperature [9]. Bulbs of *F. meleagris*, regenerated in vitro, also become dormant and cease growing, sprouting and forming leaves, if they had not been exposed to low temperature treatment [5]. Fritillary bulbs in vitro were inadequate for further morphogenesis if it was warm (24 °C) through the whole monitored period, similarly to many other geophytes [10]. Bulb regeneration in onion is well documented and controlled by day length and plant growth regulators (PGR), where gibberellic acid (GA_3_) is crucial in this process [11,12]. Recently, Yamazaki et al. [13] reported the induction of axillary bud formation and tillering in Welsh onion by the addition of GA_3_. Exogenous application of GA_3_ increased clove number per bulb and altered bulb morphology [14]. Gibberellins play an important role in regeneration of newly formed bulbs in vitro [5], lateral bud formation [15], plant morphology [14] and sugar composition [12] in many geophytes. Our previous results show that soaking in GA_3_ improved sprouting and bulb weight if snake’s head fritillary bulbs were previously grown at low temperature [5]. Growth regulators which inhibit the biosynthesis of gibberellins are known as growth retardants or gibberellin biosynthesis inhibitors (GA inhibitors). Some of these substances have N-containing heterocycle: ancymidol, flurprimidol (Flur) and paclobutrazol. They block gibberellin biosynthesis by stopping the conversion of ent-kaurene to ent-kaurenoic acid [16]. Inhibitors of GA biosynthesis decrease cell division rates, leading to more compact plants, and this ability can be used in crop production and horticulture [17].

Metabolism of sugars is linked to bulb dormancy, the beginning of sprouting and the further growth of a plant [18]. Bulbs at different developmental stages have different physiological status that can be studied [19]. Most geophytes had a high percentage of starch in their storage organs during dormancy. This carbohydrate hydrolyzed and provided energy in the form of soluble sugars during further growth. Other geophytes, like *Allium*, had bulbs with very low starch content [20]. When bulbs were previously exposed to low temperatures, starch breakdown and an increase in sucrose concentration (from hydrolysis of starch) occurred, as detected in lily [21]. Accumulation of both glucose and fructose in fritillary bulbs was significantly higher after low temperature treatment [22]. The changes in total sugar content were linked to axillary bud formation and meristem differentiation. Application of gibberellins could improve firmness and color of fruit and also its nutritional quality in the form of sugar content [14].

The aim of the study was to clarify the effect of temperature, gibberellic acid and inhibitors of GA synthesis on bulb sprouting and overcoming the dormancy of Snake’s head fritillary in vitro derived buds. The results might provide a new perspective for the improvement of fritillary bulb development, sugar distribution, growth regulation and role of GA_3_ in bulb physiology during sprouting in fritillary.

## 2. Results

### 2.1. Effect of GA_3_, GA Inhibitors and Temperature on Fresh Weight of Bulbs

After the first two weeks of culture, a significant fresh weight increase was observed when bulbs were grown on medium supplemented with GA_3_ and under standard growth conditions at 24 °C (Table 1). After two weeks of growth, there was no significant difference between the fresh weight of bulbs cultured with A and P compared to the control. Under the same growth conditions, after five weeks of culture, a significant decrease in fresh weight was observed only in bulbs grown on medium with P in comparison to the control. If we calculated the fresh weight increase after 5 weeks of growth in comparison to initial fresh weight (~90 mg/bulb), the fresh weight increase ranged from 31.1% for bulbs grown on medium with P to 57.7% for bulbs grown on medium with GA_3_.

On the contrary, when bulbs were grown at low temperature (7 °C) for two weeks, a significant increase in fresh weight was observed in bulbs cultured on both GA_3_ and GA inhibitors (A and P, Table 2.).

After five weeks of growth at 7 °C, a significant increase in fresh weight was also observed when bulbs were grown on medium with A (Table 2). Cultivation with GA_3_ for five weeks at 7 °C led to an increase in fresh weight by 18.8%, which is much lower than for bulbs grown at 24 °C. The fresh weight increase for bulbs cultured with A and P was 11.1% and 7.8%, respectively, when bulbs were grown at 7 °C.

### 2.2. Effect of GA_3_ and GA Inhibitors and Temperature on Sprouting of Bulbs

We assumed that in vitro bulbs of *F. meleagris* will significantly sprout in the presence of GA_3_ at both temperatures (Table 3 and Table 4). Bulbs cultured with GA_3_ showed no morphological differences after one week compared to the control. Spontaneous, statistically non-significant sprouting was detected when bulbs were cultured on medium with A (Figure 1A) or P (Figure 1B) at 24 °C as well as at 7 °C. This spontaneous, unexpected sprouting was noticed when bulbs were grown more than two weeks in culture on media with GA inhibitors.

Bulbs grown on medium with GA_3_ at 7 °C (Figure 1C) were small and less developed than those cultured at 24 °C (Figure 1D–F). Bulbs cultured and sprouted at 24 °C were generally higher and had more lateral bulbs and leaves (Table 3 and Table 4). Sporadically bulbs sprouted when they were cultured with A and P during cold treatment (Table 4), but a greater number was sprouted after growing at 24 °C (Table 3).

The beginning of sprouting was noticed after three weeks for control bulbs and sprouted bulbs continued to grow after five weeks. During this time, necrosis of external tissues was detected for some bulbs cultured without growth regulators or with P (Figure 1B). Some bulbs cultured with GA_3_ started to sprout after one week. Sprouted bulbs were green and continued to grow after two (Figure 1E) and five weeks (Figure 1F). There was no visible necrotic tissue over the course of time.

### 2.3. Morpho-Anatomical Analysis of the Effect of GA_3_, GA Inhibitors and Temperatures on Sprouting of F. meleagris Bulbs

In bulbs grown on medium with GA_3_ at 7 °C, both parenchyma and epidermal cells were typical for intact bulbs which were not sprouted (Figure 2A,B). Intercellular spaces were observed between cells which had usual shape. Some bulbs sprouted spontaneously and had disrupted cells of unusual shape, but in this case intercellular spaces were absent. Numerous amyloplasts were visible in both sprouted and non-sprouted bulbs, but their number decreased when sprouts became larger and visible to the naked eye. A number of meristematic regions were detected in sprouted bulbs when bulbs were cultured with GA_3_ (Figure 2B) after five weeks. Meristems were localized in the sprout. Sprouts had a distinct epidermal layer and parenchyma cells with visible nuclei. In larger sprouts, amyloplasts were less visible. Spontaneous sprouting was also detected when bulbs were cultured with GA inhibitors at 7 °C. Their anatomy is presented in Figure 2C, (paclobutrazol) and Figure 2D, (ancymidol). On both inhibitors, cells were compact and without intercellular spaces. There were no localized meristems in sprouts. Cells of spontaneously sprouted bulbs were tightly packed with a very small number of amyloplasts.

Bulbs cultured at 24 °C sprouted on medium with GA_3_ (Figure 2E,F). Cells were compact without spaces between them and had a lower number of amyloplasts. Amyloplasts were visible in some cells but not in all, as was the case for bulbs grown at 7 °C. When bulbs were cultured with inhibitors (Figure 2G,H), bulb scales were smaller and partially separated. Cuticle layer at the outer surface was less thick than in bulb scales of GA_3_-treated bulbs. Wall thickness and lignification-grade of parenchyma cells (the majority of which appeared extremely curved) were decreased compared to GA_3_-treated bulbs. Bulb scales were smaller and partially separated.

### 2.4. Effect of GA_3_ and GA Inhibitors and Temperature on Total Sugar, Fructose and Glucose Content

Soluble sugar concentrations in bulbs cultured on medium without growth regulators did not change after two weeks of culture at 24 °C, while after 5 weeks, soluble sugar content significantly decreased (Figure 3A). Soluble sugar content significantly increased in bulbs cultured with P or GA_3_ after 5 weeks at 24 °C. The highest concentration of total soluble sugars was found in bulbs grown for five weeks at 24 °C on medium with GA_3_ (Figure 2A). The addition of A led to a significant decrease in sugar concentration after two or five weeks compared to their concentration after one week. Control bulbs cultured on medium without growth regulators had higher sugar content than treated bulbs at the beginning of the experiment (after 1 or 2 weeks), but at the end, the sugar concentration dropped. Unlike bulbs at 24 °C, those cultured at 7 °C (Figure 3B), showed the highest sugar concentration when grown on control medium, without GA_3_ or inhibitors (360.71 mg/g after two weeks). Sugar concentration in bulbs treated with GA_3_ gradually increased over the course of 5 weeks, but this increase was 42% lower than in bulbs cultured at 24 °C. Bulbs cultured with A showed a constant decrease in sugar concentration during the time. Soluble sugar content in control bulbs, at both temperatures, falls into a narrow value range.

Fructose concentration in bulbs was generally low, compared to total sugars and glucose concentration, at both examined temperatures (Figure 4A,B). After one week of culture, a significant drop in fructose concentration was observed in bulbs cultured on medium with P and grown at 24 °C (Figure 4A). The highest fructose concentrations (19.94–21.81 mg/g) were detected in bulbs after two weeks on medium with A, P and GA_3_ at 24 °C (Figure 4A). Fructose concentration in control bulbs grown at 7 °C showed a constant increase. Contrary to this, bulbs cultured with GA_3_ and GA inhibitors had similar fructose concentrations after one week of cultivation at chilling temperature (Figure 4B). The lowest fructose concentration was detected with P after five weeks of culture.

A remarkable increase in glucose concentration (307% related to control) was detected in bulbs cultured on medium supplemented with GA_3_ after five weeks of growth at 24 °C (Figure 5A). Glucose concentrations in bulbs grown on media with GA inhibitors (A or P) at 24 °C showed no significant differences. Generally, lower glucose concentrations were detected when bulbs were cultured at 7 °C in all treatments in comparison to 24 °C (Figure 5B). The highest glucose concentration was recorded in bulbs cultured for one week on medium with GA_3_. Glucose concentration in bulbs grown for 2 and 5 weeks on media with either GA_3_ or GA inhibitors at 7 °C was higher compared to control bulbs.

## 3. Discussion

GA_3_ had a positive effect on the sprouting of *F. meleagris* bulbs with or without cold pretreatment, but a greater number of bulbs was noticed only when bulbs were exposed to low temperature pretreatment [5]. Without chilling, there was no positive influence on bulb multiplication and fresh weight of GA_3_ treated bulbs, which was opposite from the results of Liu et al. [14] in garlic bulbs. The authors also found that there was a difference in bulb weight among garlic cultivars. In our study, the addition of GA_3_ to solid medium strongly increased the number of sprouted bulbs and their weight with or without low temperature treatment. GA_3_ application can positively affect fruit weight [23,24], and this was exactly the case as in our study. Fruit development is an energy-demanding process, and the required energy could be supplied from sugars. Bulbs cultured with GA inhibitors had higher fresh weight than control bulbs, which indicated that GA_3_ does not play a key role in bulb biomass production during development. Generally, GA inhibitors enhance bulb size, as in Easter lily liquid culture [25], which was confirmed for fritillary bulbs compared to the control. Fritillary bulb fresh weight increased with GA_3_ in medium (even without chilling), so there was reason to suppose that the bulbs have been stressed by the absence of low temperature [26]. These stressful conditions, in combination with GA_3_ and inhibitors, led to an increase in bulb size and fresh weight. Similar stress-related changes were noticed in onion bulbs under water deficit [27]. Le-Guen-Le-Saos et al. [12] noticed a significant growth promoting effect in shallot when 10–15 μM A was added to medium. Ancymidol and paclobutrazol treated plants regenerated bulbs that were 30–60% larger than in the control, which was in accordance with our results (although in our study the observed fresh weight increase was much smaller, 7.8–38.8%, depending on the temperature regime). A smaller fresh weight increase for bulbs with inhibitors compared to GA_3_-treated bulbs was expected because plants treated with A displayed a decrease in growth and their GA content was reduced [28]. GA_3_ in shallot led to a smaller fresh weight increase in comparison to our study for snake’s head fritillary. However, the percentage of sprouting was significantly lower when bulbs were cultured with GA inhibitors, at both temperatures, than in control and especially in GA_3_-treated bulbs. This observation implied an important role of GA_3_ at the beginning of sprouting even when bulbs were not exposed to low temperature. A lower number of sprouted bulbs was detected during culturing at 7 °C, which also indicated important involvement of gibberellins in bulb dormancy release. Injecting garlic plants with GA_3_ highly promoted lateral bud formation and incidence rate in garlic [15] after 30 days of culture. The authors observed small bulbs and cloves around the main bulb, as was also found in our study for bulbs cultured with GA_3_ (Figure 1E,F). The greater number of axillary buds formed in garlic than in our study on fritillary were presumably due to in vitro culture and shorter monitoring time.

Ancymidol and paclobutrazol exhibited similar influence on bulb sprouting of snake’s head fritillary in vitro derived bulbs. Bulbs sporadically sprouted regardless of the GA inhibitors presence in medium. According to available data, inhibitors of GA synthesis stimulate the formation of bulbs and other storage organs, but our results show that some bulbs sprouted in their presence. Spontaneous sprouting of bulbs in the presence of GA inhibitors could be explained as a consequence of the fact that: (I) endogenous GA_3_ takes over a key role, (II) concentration of inhibitors was not sufficient to block sprouting, (III) used inhibitors did not have strong impact on fritillary bulbs or (IV) some other still unknown mechanism. Future experiments should be directed towards clarification of the mode and the concentration at which inhibitors need to be applied, in order to achieve total inhibition of sprouting, assuming that such total inhibition is even possible.

Sprouted bulbs with GA inhibitors and GA_3_ had almost the same morphology and growth rate at both tested temperatures. Sprouting occurred at both temperatures, but the number of sprouted bulbs was lower when bulbs were cultured with inhibitors. This result proves that GA_3_ was not the only factor that affected sprouting. In addition, control bulbs sprouted at a higher percentage than inhibitor-treated bulbs. These observations indicate that the effect of inhibitors is not linked with the entire GA synthesis inhibition in plant. We supposed that there is some endogenous mechanism which could initiate bulb sprouting, apart from exogenous GA_3_ in medium. Bulbs cultured with GA_3_ and inhibitors looked healthier and without necrotic tissue compared to non-GA_3_ bulbs. GA inhibitors are often used to stimulate bulb multiplication and reduce leaf growth as well as their damage and potential necrosis [29], which explains the adequate plant morphology when bulbs were cultured with A and P. All plants exhibited signs of necrosis after several weeks at 24 °C except those cultured with GA_3_. This observation fortified the assumption about GA_3_ involvement in the further sprouting process. However, exogenous plant hormones and their influence on endogenous hormonal balance and bulb development are poorly understood in general. Our previous study showed that bulbs in GA_3_ solution had a higher sprouting capacity [30], but the experiments were conducted after low temperature treatment and also the experiment was with bulbs soaked in GA_3_ solution. Liu et al. [14] found that exogenously added GA_3_ led to an increase in cloves number per *A. sativum* bulb, which was an efficient way to improve reproduction. The authors suggested that GA_3_ requirement was species specific and that the exact doses that improve bulb quality should be determined. We supposed that this was the same principle for GA inhibitors. In addition, the effect of inhibitors could be reduced or annulled by simultaneous GA application [28,31], which would be considered for our further experiments.

Liu et al. [15] confirmed that the soluble sugar content in garlic bulbs treated with GA_3_ was higher than in control bulbs just after four days, in almost all cultivars used. This is not fully in accordance with our results where sugar concentration increased dramatically after five weeks with GA_3_ at 24 °C. This time discrepancy can be attributed to morphogenesis in vitro, which differs from this process ex vitro. In addition, bulbs were not under chilling temperature before GA_3_ treatment, which is a requirement condition for sprouting and potentially sugar accumulation. It is reasonable to suppose that the huge increase in sugar content after five weeks of treatment with GA_3_ marks the onset of sprouting and potentially an initiation of axillary meristem. Our results show that the highest percentage of sprouted bulbs was found after exactly five weeks of culture on medium with GA_3_ at 24 °C, which correlated with the onset of meristem formation. Further investigations are needed to determine the exact time of meristem initiation after treatment with GA_3_. Considering that sugar content was lower with GA inhibitors after five weeks, we assume that GA_3_ was one of the responsible factors for sugar accumulation in bulbs. Le-Guen-Le Saos et al. [12] proved that A promotes mobilization of sugars from medium to leaves and hypothesized that high sugar concentrations in leaves resulted from the inability of onion plants to utilize glucose and fructose so rapidly. Our findings for A and P are totally the opposite. Sugar accumulation in bulbs (treated with A and P) was lower than in control bulbs grown at both temperatures. If we assume, like Le-Guen-Le Saos et al. [12], that inhibitors help sugar accumulation, we must also suppose a remarkable sugar utilization, which is unlikely. We would rather assume that inhibitors prevented (in most cases) starch hydrolysis in bulbs. Ancymidol caused higher level of soluble sugars in bulbs than P, but lower than in control plants, similar to the decrease in sucrose content in *Allium* sp. [20]. It is also known from the literature that the sensitivity to gibberellins can depend on species, even cultivars, and can elicit different responses in plants, such as different tillering capacities in different Welsh onion cultivars treated with GA_3_ [13]. Paclobutrazol in medium led to low sugar accumulation (lower than control and ancymidol-treated bulbs) at both tested temperatures. Although P blocks GA biosynthesis at the same step as ancymidol [16], bulbs treated with P showed lower sugar content. There is some evidence that there was different cell sensitivity to GA inhibitors with the same biochemical function [32]. According to this, we can postulate that fritillary cells might be more sensitive to P than to A in the manner of biochemical pathway which was responsible for sugar accumulation in bulbs. Norman et al. [33] reported that P inhibited ABA biosynthesis more strongly than ancymidol, which could be one of the reasons for the differences in sugar composition between bulbs treated with these two inhibitors. In addition, Ziv [34] concluded that P was more efficient than A in the corm formation of gladiolus. There is a possibility that certain unknown mechanisms could activate or block GA synthesis, totally unrelated to the presence of inhibitors in medium, but this mechanism does not result in changes in bulb fresh weight or sprouting.

We found that when plants were cultured at 7 °C, sugar accumulation in bulbs was lower in the presence of GA_3_ and GA inhibitors. This finding was opposite from sugar screening in bulbs grown at 24 °C. Generally, sugar concentration under chilling temperature was higher in control bulbs, compared to treated ones. Soluble carbohydrates, predominantly sucrose, have been considered as inducers of bulb formation [35] and we assumed that bulb swelling occurred under low temperature treatment. Glucose and fructose levels were lower in bulbs cultured at 7 °C, which supports the finding that sucrose was the main sugar in dormant bulbs. During bulb development at 7 °C, the lowest sugar accumulation was detected in bulbs cultured with P. This observation leads to the previous assumption about the different sensitivity of fritillary cells to two GA inhibitors with the same biochemical properties. Nevertheless, bulbs sprouted even then, and even more sprouted bulbs were noticed when they were treated with GA_3_. This observation could be explained as a proof that low temperature is a necessary precondition for bulb sprouting and further plant development and also that sugar accumulation was independent from exogenous GA_3_. GA_3_ in medium only improved sugar reserves in a bulb and its sprouting, but low temperature was the only required condition for further development. We wanted to determine sugar concentration and sugar distribution in plant after cold treatment. The sugar concentration in control bulbs cultured at 7 °C was shown to remain almost unchanged over the course of five weeks. After chilling, sugar concentration rapidly increased in the form of glucose and stayed at the same level during all five weeks. The logical next step for further experiments will be to examine sugar distribution after five weeks at 24 °C and track potential changes in sugar composition and localization.

## 4. Materials and Methods

### 4.1. Plant Material

Bulb cultures of *F. meleagris* L. were initiated according to our previously reported procedures by Petrić et al. [36]. Stock cultures were maintained on Murashige and Skoog (MS) medium (Murashige and Skoog, [37]) with 3% sucrose, 0.7% agar supplemented with 1.0 mg L^−1^ thidiazuron (TDZ) for shoots and bulbs multiplication. pH of culture medium was adjusted to 5.8 with 1 N NaOH before autoclaving. Cultures were grown in culture room at temperature 24 ± 2 °C and 16 h light/ 8h dark photoperiod with irradiance of 40 µmol m^−2^ s^−1^. Two months after the last subculture, regenerated bulbs (~80–100 mg fresh weight) were used for further experiments.

### 4.2. Culture Media and Growth Conditions

Isolated bulbs were grown on plant growth regulator free medium (MS, control) or MS media supplemented with plant growth regulators (PGR): GA_3_ or GA inhibitors ancymidol (A) and paclobutrazol (P), for 1, 2 and 5 weeks. GA_3_, A and P were sterilized by filtration through a 0.2 μm Dynagard membrane and added to medium after autoclaving to the final concentration of 10 μM. Cultures were grown at the standard culture room temperature (24 ± 1 °C) and low temperature (7 ± 1 °C) in growth chamber. An average of five bulbs were cultured in glass tubes (6 for each treatment) filled with 10 mL of culture medium. The fresh weight increase (%) was calculated as the fresh weight of bulbs after 5 weeks of treatment—initial fresh weight/ initial fresh weight × 100.

### 4.3. Morpho-Anatomical Study

Bulbs cultured for 1, 2 or 5 weeks at 7 °C were fixed in FAA fixative (5 mL of 40% formalin, 5 mL of glacial acetic acid and 90 mL of 70% ethanol) at 4 °C for 24 h, according to the procedure of Jensen [38]. Fixed material was washed, dehydrated in a graded ethanol series and embedded in paraffin. Cross sections (5–7 μm thick) were stained with hematoxylin or alcian blue, and photographed using a Leitz DMRB photomicroscope (Leica, Wetzlar, Germany).

### 4.4. Sugar Analysis

#### 4.4.1. Determination of Total Sugars Content

The total sugar content was determined according to a modified Dreywood method [39]. The procedure is based on anthrone reaction with carbohydrate in the sample which leads to a clear green color to dark green color reaction that has an absorption maximum at 620 nm. Samples (100 mg) were first boiled for 3 h with 2.5 N HCl. After cooling, Na_2_CO_3_ was added for neutralizing the reaction mixture. The neutralized material was centrifuged (5000× *g* for 2 min) and the supernatant was used for further analysis. The cooled material was mixed with a 0.2% solution of anthrone (Sigma Aldrich, St. Louis, MO, USA) in concentrated sulfuric acid (4 mL) and boiled for 10 min. After cooling, the total sugars content was determined by taking the absorbance at 620 nm (Thermo Scientific Multiskan FC, Waltham, MA, USA) using glucose as a standard.

#### 4.4.2. Fructose Determination

Fructose determination was done according to a modified Messineo and Musarra [40] protocol. The reagents used were: 75% sulphuric acid, 2.5% cysteine hydrochloride, 100 μg mL^−1^ tryptophan in 0.1 M HCl (tryptophan hydrochloride). Briefly, 2.5 mL of H_2_SO_4_ (75%) was added to 50 mg of plant material, and vortexed. Further 0.1 mL cysteine hydrochloride was added and shaken again. The mixture was placed in water bath at 45–50 °C for 10 min (green color complex). After cooling, 1 mL tryptophan hydrochloride solution was added to the mixture and the absorbance of the solution was read at 518 nm after one hour. A standard solution of fructose was used for generating the standard curve.

#### 4.4.3. Glucose Determination

Two enzymes (glucose oxidase and peroxidase) were used for glucose determination by a modified method of Amaral et al. [41]. The mixture contained 0.25 mg ortho-dianisidine in 1 mL methanol, 0.1 M phosphate buffer, peroxidase and glucose oxidase (Sigma Aldrich). One milliliter of this mixture was added to 100 mg material, centrifuged and placed in water bath at 35 °C for 40 min. Reaction products had a pink color, the intensity of which was proportional to the glucose concentration in sample. The reaction was stopped by the addition of 2 mL 6 N HCl. Absorbance was measured at 540 nm.

### 4.5. Statistical Analysis of Data

The results of all experiments are presented as mean values ± standard errors. Statistical analyses were performed using StatGrafics software version 4.2 (Statistical Graphics Corporation, Rockville, MD, USA). Data were subjected to analysis of variance (ANOVA) and comparisons between the mean values of treatments were made by the least significant difference (LSD) test calculated at the confidence level of *p* ≤ 0.05. The population of bulbs which was used in all experiments with determination of sugar concentration in bulbs treated with GA_3_ and GA inhibitors was 30 bulbs per treatment. The results reported in this study are means of three independent experiments. The graphical representation of the results was done using MS Excel.

## 5. Conclusions

The physiological and metabolic pathways in fritillary bulbs under GA_3_ treatment, and also its inhibitors, are generally unknown. This study evidenced an important role of GA_3_ in fritillary sprouting and sugar accumulation, whereas it is still unclear how exactly GA_3_ affects fritillary bulbs to start sprouting. In addition, glucose was the main sugar in fritillary bulbs. GA_3_ positively affected sugar accumulation in bulbs without low temperature treatment. In addition, bulbs treated with low temperature had higher sugar concentration than the control. Fritillary bulbs had a high percentage of soluble sugars during dormancy, as well as at the beginning of sprouting and growth. To our knowledge, this is the first report about the effect of exogenous GA_3_ and GA inhibitors on fritillary bulb morphology and sugar composition. This work provided a new insight into the bulb dormancy, reproduction and sprouting physiology of *F. meleagris*.

## Figures and Tables

**Figure 1 plants-09-01449-f001:**
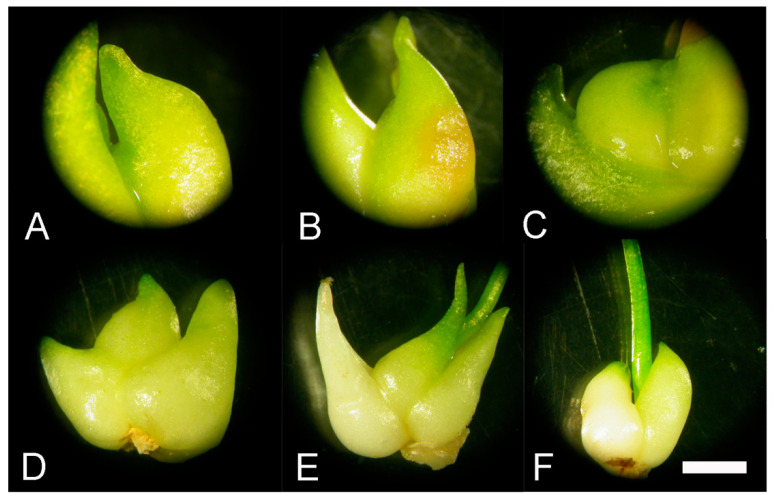
Sprouting of bulbs cultured *F. meleagris* bulbs cultured on medium with 10 μM GA_3_ and GA inhibitors. (**A**,**B**) Bulbs grown for five weeks at 24 °C on culture medium supplemented with GA_3_ inhibitors: ancymidol (**A**) and paclobutrazol (**B**). (**C**) Bulb grown for five weeks at 7 °C on culture medium supplemented with and GA_3_. (**D**-**F**) Bulbs cultured on medium with GA_3_ after one (**D**), two (**E**) and five weeks (**F**) at 24 °C. Scale bars = 5 mm.

**Figure 2 plants-09-01449-f002:**
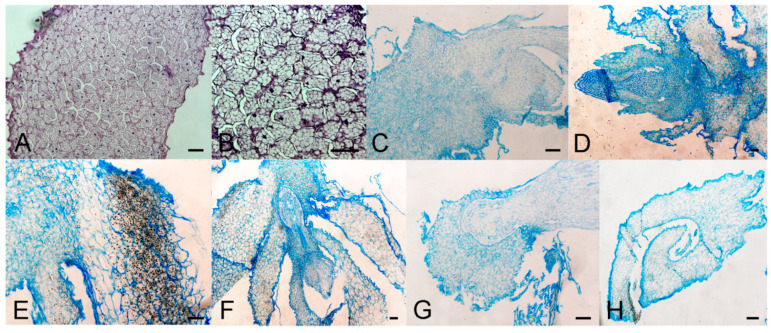
Morpho-anatomical study of *F. meleagris* bulbs sprouting after 5 weeks of culture with GA_3_ and GA inhibitors. (**A**,**B**) Cross sections of non-sprouted bulbs grown on medium with GA_3_ and cultured at 7 °C. (**C**,**D**) Cross sections of sprouted bulbs grown on medium with GA inhibitors: paclobutrazol (**C**) and ancymidol (**D**) cultured at 7 °C. (**E**–**H**) Cross sections of sprouted bulbs cultured on medium with GA_3_ (**E**,**F**), paclobutrazol (**G**) and ancymidol (**H**) at 24 °C. Scale bars = 100 μm.

**Figure 3 plants-09-01449-f003:**
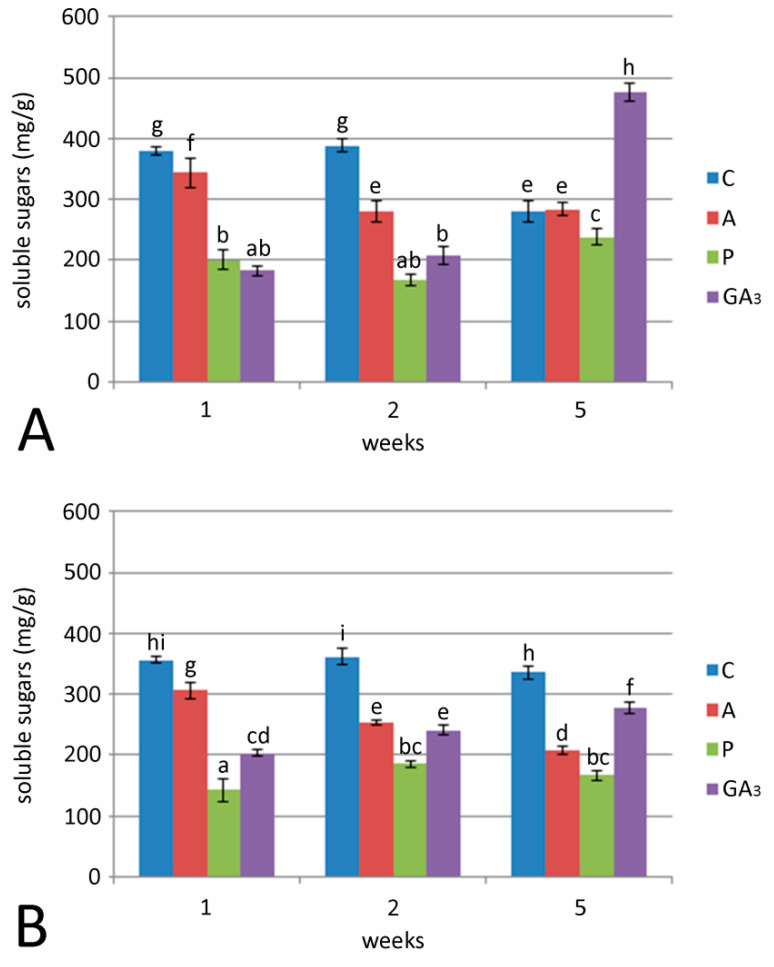
Effect of GA_3_ and GA inhibitors: ancymidol (A) and paclobutrazol (P) on soluble sugar concentration in *F. meleagris* bulbs cultured for 1, 2 and 5 weeks at 24 °C (**A**) and 7 °C (**B**). C—bulbs cultured on medium without PGR; different letters indicate significant differences between treatments at *p* < 0.05 (LSD test).

**Figure 4 plants-09-01449-f004:**
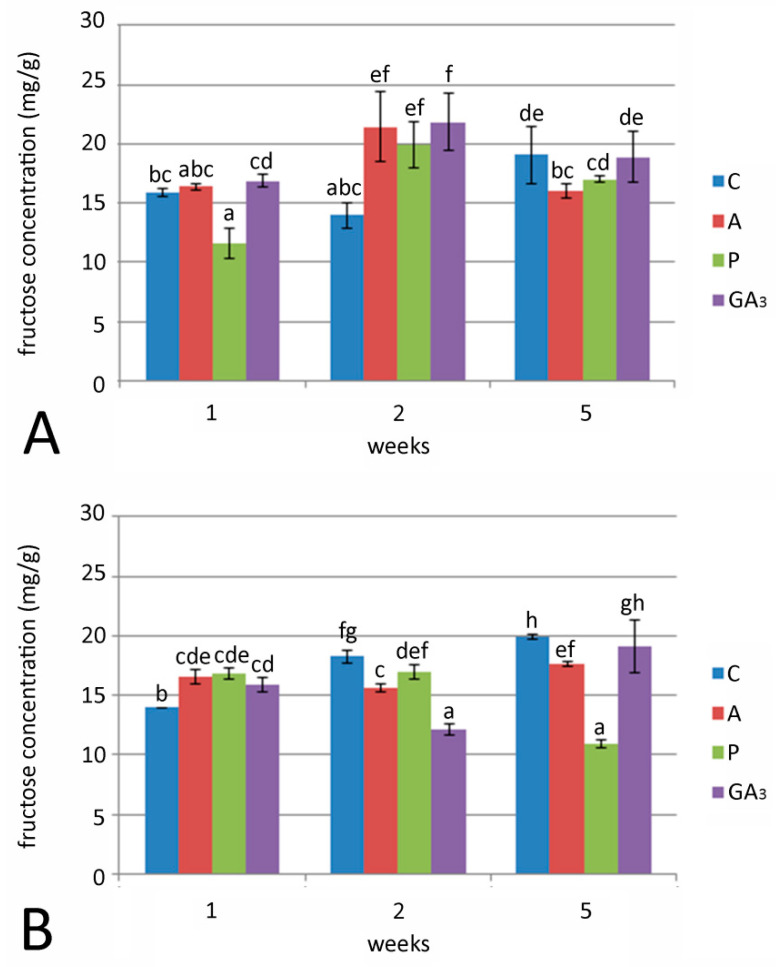
Effect of GA_3_ and GA inhibitors: ancymidol (A) and paclobutrazol (P) on fructose concentration in *F. meleagris* bulbs cultured for 1, 2 and 5 weeks at 24 °C (**A**) and 7 °C (**B**). C—bulbs cultured on medium without PGR; different letters indicate significant differences between treatments at *p* < 0.05 (LSD test).

**Figure 5 plants-09-01449-f005:**
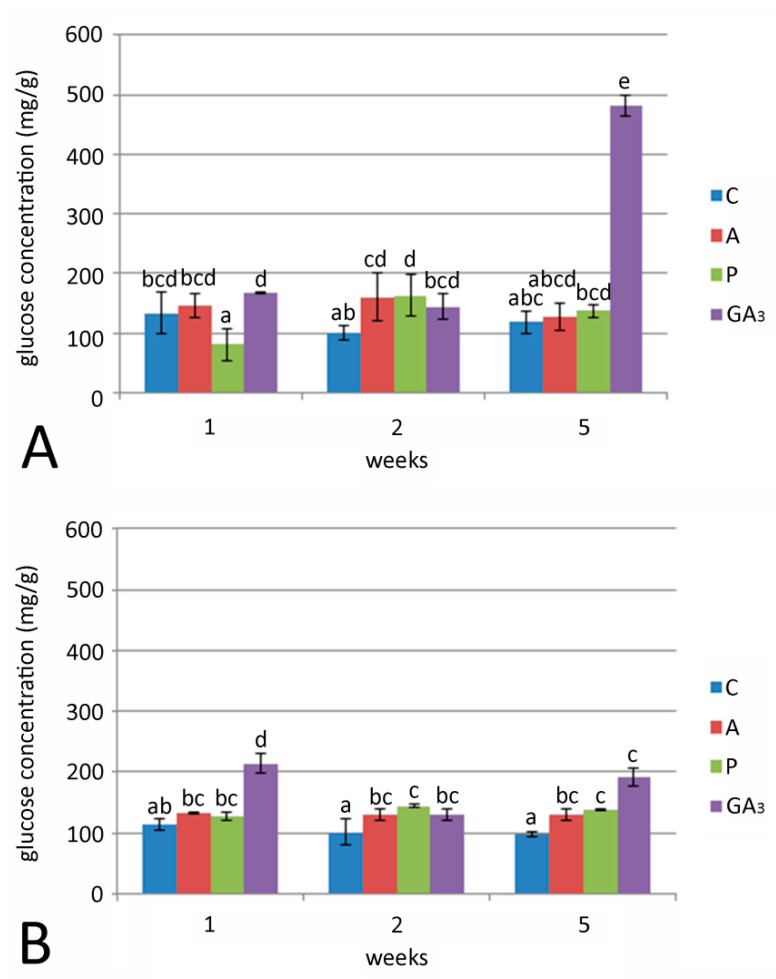
Effect of GA_3_ and GA inhibitors: ancymidol (A) and paclobutrazol (P) on glucose concentration in *F. meleagris* bulbs cultured for 1, 2 and 5 weeks at 24 °C (**A**) and 7 °C (**B**). C—bulbs cultured on medium without PGR; different letters indicate significant differences between treatments at *p* < 0.05 (LSD test).

**Table 1 plants-09-01449-t001:** Effect of gibberellic acid (GA_3_) and gibberellin biosynthesis (GA) inhibitors (10 μM): ancymidol (A) and paclobutrazol (P) on bulb fresh weight (mg) cultured for 1, 2 and 5 weeks at 24 °C.

Culture Medium	Time (Weeks)		
	1	2	5
MS	95.34 ± 2.81 ^a^*	104.13 ± 2.48 ^a^	124.47 ± 2.13 ^b^
MS + A	100.35 ± 1.39 ^ab^	104.16 ± 1.54 ^a^	125.12 ± 1.72 ^b^
MS + P	98.25 ± 1.64 ^ab^	103.16 ± 1.75 ^a^	118.06 ± 2.05 ^a^
MS + GA_3_	102.61 ± 1.89 ^b^	111.96 ± 1.73 ^b^	142.81 ± 2.06 ^c^

* Values represent average fresh weight of bulb in mg ± SE. Different letters within columns indicate significant differences between treatments at *p* < 0.05 (LSD test).

**Table 2 plants-09-01449-t002:** Effect of GA_3_ and GA inhibitors (10 μM): ancymidol (A) and paclobutrazol (P) on bulb fresh weight (mg) cultured for 1, 2 and 5 weeks at 7 °C.

Culture Medium	Time (Weeks)		
	1	2	5
MS	86.57 ± 2.02 ^a^*	88.53 ± 2.31 ^a^	92.69 ± 2.47 ^a^
MS + A	94.9 ± 1.47 ^b^	95.86 ± 1.46 ^b^	100.26 ± 1.39 ^b^
MS + P	93.2 ± 1.3 ^b^	93.96 ± 1.34 ^b^	97.46 ± 1.47 ^ab^
MS + GA_3_	90.97 ± 2.14 ^b^	96.06 ± 2.47 ^b^	107.03 ± 2.49 ^c^

* Values represent average fresh weight of bulb in mg ± SE. Different letters within columns indicate significant differences between treatments at *p* < 0.05 (LSD test).

**Table 3 plants-09-01449-t003:** Effect of GA_3_ and GA inhibitors (10 μM): ancymidol (A) and paclobutrazol (P) on bulb sprouting (%) cultured for 1, 2 and 5 weeks at 24 °C.

Culture Medium	Time (Weeks)		
	1	2	5
MS	0.00 ± 0.00 ^a^*	6.44 ± 1.86 ^b^	8.32 ± 2.64 ^a^
MS + A	0.00 ± 0.00 ^a^	0.00 ± 0.00 ^a^	7.52 ± 1.07 ^a^
MS + P	0.00 ± 0.00 ^a^	1.07 ± 1.07 ^a^	6.45 ± 0.00 ^a^
MS + GA_3_	11.82 ± 1.07 ^b^	12.09 ± 0.00 ^c^	29.02 ± 1.86 ^b^

* Values represent average percentage of sprouted bulbs (%) ± SE. Different letters within columns indicate significant differences between treatments at *p* < 0.05 (LSD test).

**Table 4 plants-09-01449-t004:** Effect of GA_3_ and GA inhibitors (10 μM): ancymidol (A) and paclobutrazol (P) on bulb sprouting (%) cultured for 1, 2 and 5 weeks at 7 °C.

Culture Medium	Time (Weeks)		
	1	2	5
MS	0.00 ± 0.00 ^a^*	0.00 ± 0.00 ^a^	5.37 ± 1.07 ^a^
MS + A	0.00 ± 0.00 ^a^	1.07 ±1.07 ^a^	1.01 ± 1.07 ^a^
MS + P	0.00 ± 0.00 ^a^	0.00 ± 0.00 ^a^	2.00 ± 2.00 ^a^
MS + GA_3_	8.59 ± 1.07 ^b^	18.27 ± 2.15 ^b^	19.35 ± 1.86 ^b^

* Values represent average percentage of sprouted bulbs (%) ± SE. Different letters within columns indicate significant differences between treatments at *p* < 0.05 (LSD test).

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
