# Peer review of "Breaking the Dormancy of Snake’s Head Fritillary (*Fritillaria meleagris* L.) In Vitro Bulbs—Part 1: Effect of GA_3_, GA Inhibitors and Temperature on Fresh Weight, Sprouting and Sugar Content"

_plants, 2020, doi:10.3390/plants9111449_

Round 1
Reviewer 1 Report
The manuscript entitled “Breaking the dormancy of snake’s head fritillary (Fritillaria meleagris L.) in vitro bulbs – Part 1: Effect of GA3, GA inhibitors and temperature on fresh weight, sprouting and sugar content” contains information which is of both practical interest and scientific value. Therefore I recommend it to be published in the Plants Journal after addressing some issues, contained in the attached pdf file.

Author Response
Response to Reviewer 1 Comments
Thank you very much for reading the manuscript. Your comments and suggestions are very much appreciated, as they helped us improve the text. All suggestions made by the reviewer were accepted, and modifications are clearly visible in the track changes version of the revised manuscript 964450. Comments concerning some of those modifications are listed below.
35 Reference needed
Appropriate reference was introduced [line 35 in revised version]
Figure 1. Provide bar
Bars were provided, and their size given in legend to the Figure 1 [line 136 in revised version]
Figure 2. Provide bar.
Bars were provided, and their size given in legend to the Figure 2 [line 136 in revised version]
172 part of the sentence is not true.....rephrase and clarify...it is important to compare to control and then between treatments
The sentence was rephrased [lines 175-176 in revised version]
176 compared to what? 1st week I suppose...rephrase
Statement was corrected [lines 180-181 in revised version]
230-232 Rephrase the sentence and clarify better
The sentence was rephrased for clarity [lines 238-241 in revised version]
269-272 redundant...it was already stated
Redundant statement was deleted.
291 redundant
Redundant statement was deleted.
Sincerely,
Authors

Reviewer 2 Report
In this study, the authors tried to clarify the effect of temperature, gibberellic acid and two inhibitors of GA synthesis on bulb sprouting capacity, fresh weight and soluble sugars content. The study is continuation of the authors’ scientific interest on breaking dormancy in snake’s head fritillary, a commercially interesting ornamental plant species. Experimentally this work was well done, the results well discussed and the work would be valuable contribution to the mentioned research field. Generally, the manuscript is nicely written, though the sections Results and Discussion could be more “condensed” i.e. shorter and less descriptive. There are some mistakes and omissions throughout the text, which has to be corrected (visible in the corrected PDF).

Author Response
Response to Reviewer 2 Comments
Thank you very much for reading the manuscript plants - 964450. Your comments and suggestions are very much appreciated, as they helped us improve the text. All suggestions made by the reviewer were accepted, and modifications are clearly visible in the track changes version of the revised manuscript 964450. Comments concerning some of those modifications are given below.
Figure 3 Please write "Soluble sugars (mg/g)" on the ordinate axis instead of "concentration (mg/g)"
Figure 4 Please write "Fructose concentration (mg/g)" on the ordinate axis instead of "concentration (mg/g)"
Figure 5 Please, write "Glucose concentration (mg/g)" on the ordinate axis.
All ordinate axes were changed according to the reviewer’s instructions. Figures 3, 4 and 5 in the text were replaced by the modified ones.
267 Please re-work the expression.
The expression was rephrased [line 276 in revised version].
Sincerely,
Authors
